# The Effect of Specimen Size on Acoustic Emission Parameters and Approximate Position of Defects Obtained during Destructive Testing of Cementitious and Alkali-Activated Degraded Fine-Grained Materials

**DOI:** 10.3390/ma16093527

**Published:** 2023-05-04

**Authors:** Libor Topolář, Dalibor Kocáb, Petr Hrubý, Luboš Jakubka, Michaela Hoduláková, Romana Halamová

**Affiliations:** 1Faculty of Civil Engineering, Brno University of Technology, Veveří 331/95, 60200 Brno, Czech Republic; dalibor.kocab@vutbr.cz (D.K.); lubos.jakubka@vutbr.cz (L.J.); hodulakova.m@fce.vutbr.cz (M.H.); romana.halamova@vutbr.cz (R.H.); 2Faculty of Chemistry, Brno University of Technology, Purkyňova 464, 61200 Brno, Czech Republic; xchrubyp@fch.vut.cz

**Keywords:** size effect, degradation, acoustic emission method, static modulus of elasticity, Kaiser effect, blast furnace slag, blast furnace cement, linear localisation

## Abstract

Two sizes of test samples were selected to investigate the effect of size on the level of degradation. The smaller test specimens had dimensions of 40 × 40 × 160 mm, and the larger ones had dimensions of 100 × 100 × 400 mm. Both sizes of test specimens were always made of the same mortar. In one case, Blast Furnace Cement was chosen as the binder. In the other case, it was an alkali-activated material as a possibly more environmentally economical substitute. Both types of material were deposited in three degrading solutions: magnesium sulphate, ammonium nitrate and acetic acid. The reference set was stored in a water bath. After six months in the degradation solutions, a static elastic modulus was determined for the specimens during this test, and the acoustic emission was measured. Acoustic emission parameters were evaluated: the number of hits, the amplitude magnitude and a slope from the amplitude magnitude versus time (this slope should correspond to the Kaiser effect). For most of the parameters studied, the size effect was more evident for the more degraded specimens, i.e., those placed in aggressive solutions. The approximate location of emerging defects was also determined using linear localisation for smaller specimens where the degradation effect was more significant. In more aggressive environments (acetic acid, ammonium nitrate), the higher resistance of materials based on alkaline-activated slag was more evident, even in the case of larger test bodies. The experiments show that the acoustic emission results agree with the results of the static modulus of elasticity.

## 1. Introduction

Thanks to its versatility, availability, quite sufficient durability, relatively low price and easy handling, concrete has been and will continue to be the most common construction material around the world. However, concrete production is associated with a large consumption of natural resources and energy, as well as with significant CO_2_ emissions, particularly due to the presence of cement as essential concrete component. It is estimated that up to one ton of CO_2_ is released during the production of one ton of Portland cement [1], and according to Statista, Inc. (Berlin, Germany) [2] 4.4 billion metric tons of cement were produced in 2021. Generally, cement production corresponds to 5 to 8% of global CO_2_ emissions [3]. Therefore, efforts to replace cement partially or fully by industrial by-products [4,5,6] and waste alumino-silicate sources such as slags and fly ashes have been increasing. Alongside other alternative binders such as calcium sulfoaluminate cements or supersulfated cements [7], alkali-activated cementless binders have attracted increasing scientific attention during the last decades.

Most building structures are subjected to compressive, tensile, moment loads and shear loads during their service life—and other external degrading influences. For example, huge amounts of concrete based on ordinary Portland cement have been used as construction materials for sewers and wastewater treatment plants. Exposure to an aggressive, acid-rich environment leads to severe degradation [8,9]. For these reasons, it is necessary to look for materials that resist degradation better than materials based on the ordinary Portland cement. Additionally, while cement is still the most commonly used binder, alternatives in the form of alkali-activated materials are emerging [10].

Alkali-activated materials are based on suitable aluminosilicate precursors with a sufficiently high content of amorphous phase, such as granulated blast furnace slag, fly ash and metakaolin. In short, after the dissolution of the amorphous phase, which is facilitated by the presence of an alkaline activator, usually sodium waterglass or sodium hydroxide solution, polycondensation reactions take place and hydration products with binding properties are formed. The binder structure and composition depend on the chemistry of both aluminosilicate precursors and activators, as well as on the curing conditions and other parameters. It can be summarised that the main hydration product of blast furnace slag alkaline activation is calcium–aluminium–silicate–hydrate, as a derivate of the C-S-H known from hydration Portland cement systems, while the alkaline activation of fly ash and metakaolin forms a more cross-linked three-dimensional structure of sodium–aluminium–silicate–hydrate [11]. To reach sufficiently high early age strength, metakaolin and fly ash usually require a high amount of activator and heat curing, which means additional economic and technological demands. That is why alkali-activated materials based on the Blast Furnace Slag, alkali-activated using the mixture of sodium water glass and sodium hydroxide, were used in this study.

Most of the recent studies dealing with the chemical degradation of cement-based material are limited to one size of test specimens, e.g., [12,13,14,15,16]. Even studies on the chemical degradation of alkali-activated materials are usually performed on small specimens of the same size within a given piece of research, e.g., [17,18,19]. In our case, we focused on the effect of degradation on different specimen sizes for two mortars on the base of CEM III/A and alkali-activated material. Those materials were chosen due to their similar industrial use and comparable properties.

Size effect is widely discussed for commonly used concretes (reinforced or unreinforced) [20,21] in terms of fracture mechanics [22,23] or theoretical calculations or models [24,25]. In other works, for example [26], researchers have tested different sizes of concrete block specimens for compressive strength under dynamic uniaxial loads.

Very few studies have evaluated the effect of sample size on degraded specimens using simultaneous non-destructive methods. However, in the case of laboratory studies using small specimens (in civil engineering, different types of pastes), researchers can make significant inaccuracies in their conclusions. Using larger specimens increases the defect risk, allowing the degradant to penetrate more easily. Thus, usually, better results (i.e., compressive strength, etc.) are obtained thanks to the use of small specimens with dimensions such as 20 × 20 × 100 mm. Our research also involves using the same mixtures for small and large test specimens, which is not commonly found in the literature. This way, the adverse effects of using different types and amounts of aggregate in the mixture were eliminated.

Another aspect that significantly distinguishes this paper from many others is the use of the acoustic emission method to reveal the effect of the size of the degraded specimens on the parameters obtained. The acoustic emission method was applied during the static modulus of elasticity test. This paper, therefore, looks at the effect of the size of the test specimens on the parameters obtained during destructive testing.

## 2. Materials

Alkali-activated blast furnace slag binder systems were prepared using slag with a Blaine fineness of 400 m^2^∙kg^−1^ (ArcelorMittal Ostrava, s.r.o., Czech Republic). The slag’s phase composition was determined by X-ray powder diffraction (XRD) using the Rietveld method with internal standard (CaF_2_). The amorphous content of slag was about 70.5%. The main crystalline phases were akermanite (19.8%), calcite (6.7%), quartz (2.7%), and merwinite (0.4%). Liquid sodium water glass—silicate modulus (Ms) = 0.5 (Vodní Sklo, a.s., Czech Republic)—was used as an alkaline activator in the 6% Na_2_O-related dose per the slag weight. The comparative cementitious binder was prepared using the CEM III/A 32.5 R (Horné Srnie cement plant, Cemmac s.r.o., Slovakia). The chemical compositions of the Blast Furnace Slag and Blast Furnace Cement were determined using X-ray fluorescence analysis (XRF), as shown in Table 1.

### Preparation and Composition of Testing Specimens—Mortars

The testing specimens were prepared as prescribed in the European standard EN 196-1; the composition for each mixture is in Table 2. The mixing took 3 min, and the activator and the slag (or CEM III/A) were mixed using a low speed during the initial 30 s, and then the standardised sand was added and mixed for another 30 s. This was followed by high-speed mixing for 30 s and continued by wiping the walls of the container for 30 s. A final mixing using high speed for 60 s was implemented at the end. The laboratory mixer (KitchenAid Robot Artisan 175) was used for the mortar preparation. The prepared mixture was cast into the moulds and compacted on the compacting table for 30 s to eliminate the air entrapped. The testing specimens were demoulded after 24 h and put into water storage, where they were stored for 28 days at 25 °C (the specimens were not stored in a tempered room; the temperature of 25 °C was the long-term average for the period under study). Then, the degradation testing started with the immersion of the testing specimens in the specific solutions (magnesium sulphate (50 g/L), ammonium nitrate (6 mol/dm^3^) and acetic acid (pH = 3), tap water as a reference). The sand-to-binder mass ratio in mortars was 3:1. The water-to-binder ratio (w/b) was the same (0.45) for both tested binders. Cement (CEM III/A 32.5R) was chosen as the binder because it was blast-furnaced according to the marking and had minimum admixtures. Blast-furnace slag is the most commonly used cement replacement in the Czech Republic’s cement-free mixes.

## 3. Testing Methods

### 3.1. Description of Degradation Environments

Degradation environments were designed to simulate various degradation mechanisms and provide reference specimens as a blank. Water embedding of specimens was chosen as a reference. A solution of acetic acid of pH ≈ 3 was selected to simulate behaviour under the acidic conditions, thus the decalcification mechanism along with the partial dissolution of the silicon-containing gels. Next, decalcification without the aligned dissolution of silicate units was studied using the 6 mol/dm^3^ ammonium nitrate solution. Sulphate corrosion was then simulated using the 50 g/L solutions of magnesium sulphate. Degradation solutions were fully replaced every 28 days to stimulate the degradation process. The pH of the acetic acid environment was measured and kept at the value of approximately pH ≈ 3 by adding an extra dose of acetic acid three times per week.

The ongoing processes were observed by measuring the pH (Mettler Toledo SevenCompact™ Duo S213 using an InLab Routine electrode with range of pH 0–14) of all the solutions before replacement with fresh ones. The progress in time can be seen in Table 3. It can be seen from the table that even though one day the pH was reduced to a value close to 3 (for acetic acid), just before the exchange the value was higher again.

Each binder type’s specimens (3 pcs of 100 × 100 × 400 mm and 21 pcs of 40 × 40 × 160 mm) were placed into one container for a specific solution (approx. 50 L). The volume ratio between the specimens and the degradation solution was 1:3 during the testing period. Containers were kept at laboratory conditions (20 °C) for the whole testing period. The effect of degradation media was intensified using the air stream bubbling.

### 3.2. Static Modulus of Elasticity and Compressive Strength

After removing the test specimens from the individual degradation solutions, the static modulus of elasticity and then the prismatic compressive strength were determined for all specimens. The modulus of elasticity test was performed according to ISO 1920-10. The loading progress of the test specimens is shown in Figure 1. The basic load stress was always 0.5 MPa (the applied force was 1 kN for smaller specimens and 5 kN for bigger specimens), and the upper load stress was 1/3 of the expected compressive strength. The compressive strength was estimated based on the results of parallel tests performed on other specimens. Still, these tests are not described in this paper (these were the flexural tensile strength, the compressive strength on fragments and the dynamic modulus of elasticity). The loading of smaller specimens was carried out in the DELTA 3-600 (FORM+TEST) test press and the loading of bigger specimens in the ALPHA 3-3000S (FORM+TEST) test press, in both cases using the Proteus software. The deformations of the test specimens were measured during testing using two electronic strain transducers LD-DD1-2 (FORM+TEST). Using the Spider8 data logger (HBM), a record of the applied force and deformation of the test specimens was obtained throughout the test. After the modulus of elasticity test, compressive strength was determined on all test specimens according to EN 12390-3.

### 3.3. Acoustic Emission Method

An acoustic emission method is a tool for the non-destructive monitoring of active dynamic changes in a stressed material [27]. The basis of this method is the continuous monitoring of the acoustic response caused by the initiation and propagation of damage under stress (mechanical, chemical or thermal) [28,29,30]. A transient elastic wave is generated locally due to the rapid release of energy as the stressed material propagates through the material until it reaches the surface. The surface waves are then captured by piezoelectric sensors and subsequently converted to electrical signals [31]. The electrical signals are amplified, processed and finally stored in a recording device. Unlike other non-destructive detection methods, acoustic emission signals can only be detected when damage occurs within the material [32].

Commonly used acoustic emission signal characteristics are shown in hit form in Figure 2, including amplitude, energy, number of rings, rise time and duration. The acoustic emission amplitude corresponds to the point of maximum value, expressed on a decibel scale. The energy of the waveform is the area above the threshold and below the envelope curve. Any oscillation of the electrical signal exceeding a threshold value is considered a ring count. The ring count can reflect the signal strength and frequency and is widely used to evaluate acoustic emission activities. The energy and amplitude of acoustic emission can be used to characterise the magnitude of damage [33,34]. The time between the start time of a single acoustic emission signal and the peak amplitude is called the rise time, and the time between the start time of a single acoustic emission signal and the decay time is called the duration [35]. In this study, only two of the above parameters were used: the number of recorded hits and the AE amplitude.

The Kaiser effect principle appeared during the specimens’ cyclic loading (see Figure 3). The AE signals were insignificant unless the stress exceeded the previously applied maximum pressure. From point A to B, the AE signals were released continuously, but after unloading up to point C and reloading until point B, no signal could be observed unless it exceeded point B. This phenomenon is called irreversibility and is known as the Kaiser effect. As long as the loading continued, the AE signals were emitted. When the loading cycles reached higher stress in point D, the material entered an unstable phase in which the previous microcracks expanded significantly and serious damage occurred. AE signals could be seen even before reaching point D, which means that the Kaiser effect at the higher stress level tends to decrease [37,38].

Linear (one-dimensional) localisation of acoustic sources was performed considering the wave propagation velocity for each material type separately (determined from ultrasonic waves); see Figure 4. Localised AE events were recorded along the line between the transducers and used for subsequent AE analysis to exclude any other received signals [39].

In the experiments presented here, the monitoring of AE activity was performed with a dual-channel unit DAKEL ZEDO (ZD Rpety, Prague, Czech Republic) with the following input parameters of hit-detector:The threshold value for individual AE hits was set at 200% above the noise level;A sampling of AE hits was set to 5 MHz;The cut-off frequency of the low-pass filter was set to 500 kHz.

The total gain was 34 dB, but only by pre-amplifier. The AE sensors were attached to the specimens with beeswax in a thin layer; see Figure 5. The location of the acoustic emission sensors on the specimen was proportional to the size of the test specimen.

## 4. The Ongoing Chemical Processes

The degradation of both tested binder systems due to the immersion in ammonium nitrate and acetic acid was related to decalcification. Decalcification represents a process where the simultaneous leaching of Ca^2+^ from the binder gel contemporaneously with the formation of corresponding calcium salts occurs. This is related to the simultaneous reduction of the C/S ratio in the binder gel, which can result in the residual S-H gel with limited utility value instead of the C-S-H or the C-A-S-H. The alkali-activated blast furnace slag has higher resistance against decalcification compared to Ordinary Portland Cement, as stated in studies such as [40,41,42]. The alkali-activated blast furnace slag binder gel has more intensely cross-linked silicate chains, since alumina units are implemented in the binder gel structure (C-A-S-H) compared to the main hydration product of cement hydration (C-S-H gel). Moreover, the A-S-H can form a stable passivation layer. Next, the pH of the pore solution of the alkali-activated slag is higher than the one in cementitious systems. Decalcification due to the effect of the ammonium nitrate takes place only when the pH of the solution is lower than 9.25 [43], so since the initial pH of the alkali-activated system was higher, it helped to prevent degradation.

Finally, there was no portlandite in alkali-activated slag. Portlandite present in a hydrated cementitious system can be easily leached out from the matrix. This protects the C-S-H gel from decalcification, but simultaneously, once the portlandite is leached, the pores of the matrix are opened and the rate of degradation is increased, since the permeability of the matrix increases.

The bigger the specimen, the lower deterioration of properties was observed. This can be explained via the mass ratio between the degradation medium and the specimen size. Bigger specimens contain more Ca^2+^ in proportion to relevant anions in the degradation solution; thus, degradation does not reach such a rate (volume and mass ratio between the specimen/liquid differs for bigger and smaller specimens). Then, aggressive solutions do not penetrate to such a depth, and the residual effective non-affected cross-section of the specimen is more significant. This effect was more severe for the cementitious system.

No degradation effect of immersion on the tap water was observed, regardless of the specimen size and the type of the binder. The immersion in the 50 g/L solution of magnesium sulphate did not result in any significant changes in the properties of the cementitious binder, no matter the size of the specimen.

The degradation of the calcium–silicate-based binders due to the effect of magnesium sulphate solution can be explained by the formation of the expansive product as a product of the reaction between the calcium ions in the matrix and sulphur anions of the penetrating solution. These reaction products can be gypsum, ettringite and brucite. The decalcification can be observed along with those processes, leaving the partially decalcified binder gel behind [44].

The small testing specimens prepared from alkali-activated slag mortar showed worse resistance against sulphate corrosion compared to the cementitious one. There is a discrepancy in the studies comparing degradation resistance between ordinary cement and alkali-activated slag.

Beltrame et al. and Mithun et al. state a lower resistance of alkali-activated binders compared to cementitious ones; on the other hand, others such as Allahvedi et al. and Bakharev et al. presented a higher resistance. The larger specimens of alkali-activated mortar did not show such rapid deterioration of tested properties as the smaller ones. This could, again, be related to the affected area of the specimen and the overall ratio between the specimen and the aggressive media [44,45,46,47].

## 5. Results and Discussion

### 5.1. Size Effect

The graphs in Figure 6 show two representative stress load curves for determining the static modulus of elasticity (red curve). The plots also show the hit record of the acoustic emission in the form of amplitude (blue points). These points are interlaced with a line (linear regression) whose slope determines the influence of the Kaiser effect [48,49,50] on the acoustic emission signals. The higher slope of this line reflects the more significant demonstration of the Kaiser effect. In the material that exhibits an initial failure under a specific load, the Kaiser effect describes the absence of acoustic emission until that load is exceeded. The Kaiser effect results from discontinuities created in the material during previous steps that do not move, expand, or propagate until the former stress is exceeded.

The graphs are just a demonstration of the two selected specimens. In the other parts of the results, the individual sets will be represented by the averages of the values and the coefficient of variation, see Table 4 and Table 5.

To demonstrate the effect of specimen size (Figure 7, Figure 8, Figure 9, Figure 10 and Figure 11), the ratio of the average values of the observed parameters was chosen when the average value of larger specimens was divided by the average value of smaller specimens. The graphs show a line segment at value one for the given ratios (red dashed line). If the ratio is close to one or exactly one, the effect of specimen size is negligible or none for the parameter of interest. If the ratio of the average values is significantly greater than one, it shows that the size effect on the parameter favours larger specimens. Similarly, if the ratio of the average values is less than one, it shows that the effect of specimen size on the parameter of interest favours smaller specimens. In all graphs, the grey bars are for mortar made from CEM III/A (CEM), and the green bars are for alkali-activated mortar.

From the graphs in Figure 7 and Figure 8, it can be seen that the size effect for the specimens immersed in water and magnesium sulphate was none (CEM) or only minor (AAS). The situation is different for the specimens immersed in ammonium nitrate and acetic acid. The cement mortar was degraded to a depth of approximately 10 mm (Figure 12 (left)), which was reflected both in the decrease in mechanical properties in general (Table 4 and Table 5) and in the different results of large and small specimens. A significantly higher decrease in values E_c_ and f_c,prism_ was observed for smaller specimens (compared to the reference specimens deposited at the point) than for larger specimens. The difference was approximately twofold for the elastic modulus and more than threefold for the compressive strength. A decrease in the mechanical properties of specimens immersed in ammonium nitrate and acetic acid compared to specimens in water was also observed for AAS mortars. However, the difference was smaller than for cement mortar; see Table 4 and Table 5. At the same time, the size effect of AAS mortar was significantly smaller than that of cementitious specimens. Again, higher mechanical properties were determined for larger specimens than smaller specimens, but the ratio was at most 1.5 times higher. This was due to the better resistance of the AAS mortar to degradation effects, as damage in the alkali-activated specimens was found to a maximum depth of 2 mm (Figure 12 (right)).

In the case of the number of recorded hits of acoustic emission, the ratio of the average values is almost always less than one. In the case of CEM mortars, the specimens could be divided into two groups. In the first group were specimens immersed in water and magnesium sulphate. In this case, there was less damage, but at the same time, a more significant attenuation of the acoustic emission signals in the case of larger specimens. In the second group were specimens immersed in ammonium nitrate and acetic acid. In the second group, the same effect was seen, but the larger specimens were already degraded and the ratio was closer to one. As a result of this higher rate of degradation, there was more damage to the material throughout the volume during the load test. A greater damage rate also means more significant acoustic emission activity; therefore, the ratio values were close to one. In the case of AAS mortars, the differences in the ratios of the average values were minimal. Therefore, it was evident that the degradation rate was also lower than in the case of CEM mortars. Thus, the size effect in the case of acoustic emission hit counts was noticeable, especially in the loss of part of the signals. This loss was due to the same gain setting for both specimen sizes and the long path of the signals in the material. The acoustic emission signal travels from the acoustic emission source to the acoustic emission sensor on the surface of the specimen.

It can be seen from the ratio of the average values of the amplitude of the acoustic emission hits that the size effect does not have a significant impact on this parameter of the acoustic emission. In the case of the CEM mortars, the ratio of the average values was close to value one. A more significant difference was observed for the specimens immersed in acetic acid solution, where the recorded amplitude was higher for the larger specimens. This manifestation was due to the greater degree of degradation of the smaller specimens, where the material was no longer cohesive enough to form significant damage that would manifest itself in a higher amplitude value.

The slope of the line determined from the amplitude of the acoustic emission hits recorded during the static modulus test should be the most telling of the acoustic emission parameters about the effect of degradation on the size effect. In the case of a healthy undegraded material, the damage should follow the Kaiser effect, and thus the slope of the line-value directive should be as significant as possible. From both graphs, the values of the slope differed the most in the case of ammonium nitrate and acetic acid solutions. In these cases, the influence of the size effect on the obtained parameter was most evident. As the degradation progressed from the surface to the depth of the test specimens, the larger specimens retained their cohesiveness for longer in a more significant part of their envelope, which was represented by a higher value of the average slope values. Again, a more pronounced degradation effect on the CEM mortar was seen in a larger ratio of average slope values.

### 5.2. Linear Localisation of Hits for Smaller Specimens

The following graphs (Figure 13, Figure 14, Figure 15 and Figure 16) show the localised events (for the three test specimens when each specimen has one colour/shape on the figure) of acoustic emission between sensors that were 0.1 m apart. Different behaviour between the different degradation environments can be seen. In the case of CEM mortars in water and magnesium sulphate, the expected behaviour occurred where events were localised throughout the area between the sensors during mechanical loading, indicating minimal or no degradation due to the degradation process. However, in the case of CEM mortars in ammonium nitrate and acetic acid, the localisation of acoustic emission events during mechanical loading was significantly different. Due to the disturbance of the material structure, the overall number of recorded hits reduced considerably, and thus the localised events were significantly less. Additionally, in the case of the specimens immersed in acetic acid, it can be seen that most of the localised events were only near the sensors, i.e., near the compression surfaces of the press. Thus, it can be assumed that the specimens experienced crushing rather than cracking within the structure.

For AAS mortars immersed in water and magnesium sulphate solution, the expected situation occurred, i.e., localised events during mechanical loading were, again, in the entire area between the sensors, indicating minimal or no degradation. The problem was very different in the case of the immersion of AAS mortars in ammonium nitrate and acetic acid solutions. There were a large number of localised events, and due to the slight decrease in the static modulus a large amount of damage was likely generated within the structure.

## 6. Conclusions

The size effect was studied using two sets of degraded testing specimens within this study. Blast Furnace slag cement—CEMIII/A (CEM)—and alkali-activated blast furnace slag (AAS) were used as the two tested binders. Both were used as mortars with the same water-to-binder ratio, and sets consisted of specimens of dimensions 100 × 100 × 400 mm and 40 × 40 × 160 mm to study the size effect. Testing specimens were immersed in aggressive media—solutions of ammonium nitrate, acetic acid, magnesium sulphate and tap water as a reference—for six months. Containers with the specimens and various solutions were bubbled via the air stream to stimulate the degradation processes. Degradation solutions were renewed every 28 days for the same purpose. Specimens were pulled out of the solutions, dried out and tested to determine the static modulus of elasticity while measuring acoustic emissions after six months of degradation (seven months of hydration in total).

AAS specimens showed a higher resistance to decalcification caused by acetic acid and ammonium nitrate solutions than CEM specimens;The effect of magnesium sulphate and tap water on measured parameters was negligible;The size effect was more significant for CEM specimens.

From the localised AE hit simultaneously with the static modulus of elasticity, when both CEM and AAS specimens were immersed in water and magnesium sulphate, respectively, there was no significant structural disruption due to degradation. This was not the case when the specimens were immersed in ammonium nitrate and acetic acid, where, particularly in the case of CEM specimens, the material’s internal structure was completely broken. Additionally, then, when loaded, this specimen tended to be crushed. In the AAS specimens’ case, only a slight structure disruption occurred due to degradation. Therefore, the localised events were more numerous, with a larger average amplitude, indicating a more cohesive material structure after degradation.

This study confirmed that the specimen size could distort the results obtained, especially when studying the degradation processes. A proper selection of the size of the test specimens when designing the experiment is crucial for the relevant determination of the service life of structural segments. It is necessary to think in a wider context, i.e., to reflect the character of the final product or construction. Catastrophic changes can be detected for small specimens, while for large specimens the degradation may be only at the surface and therefore does not necessarily lead to the failure of the entire structure.

## Figures and Tables

**Figure 1 materials-16-03527-f001:**
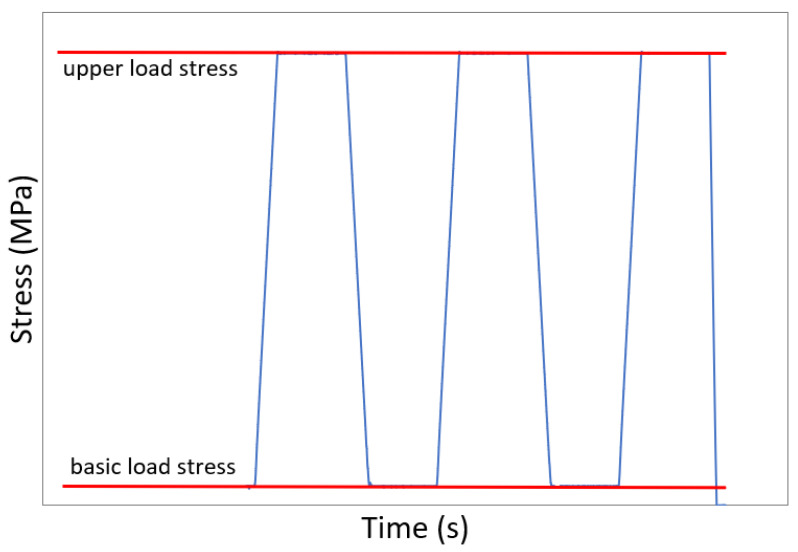
The loading progress of the test specimens.

**Figure 2 materials-16-03527-f002:**
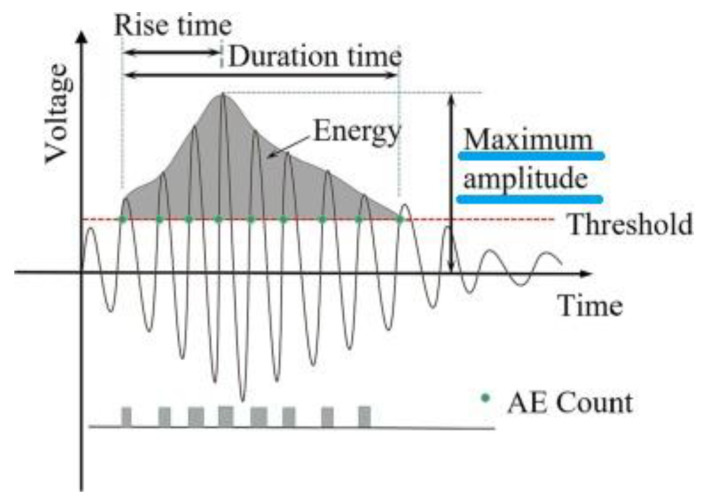
Schematic diagram of AE parameters [36].

**Figure 3 materials-16-03527-f003:**
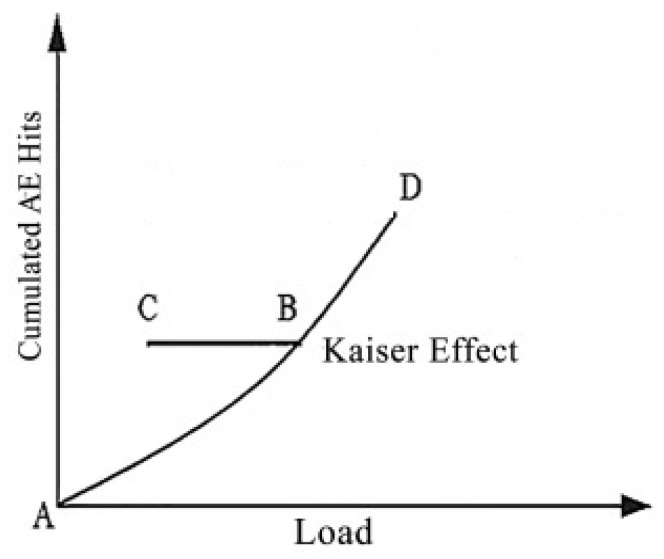
Principle of Kaiser effect.

**Figure 4 materials-16-03527-f004:**
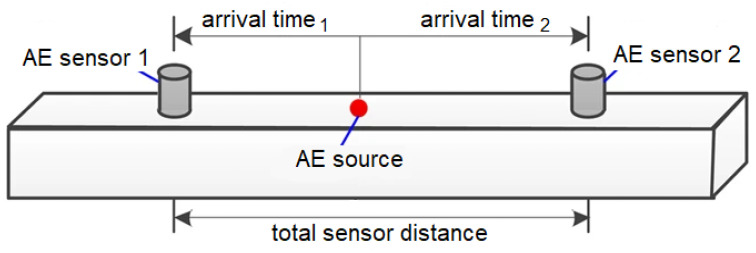
Schematic diagram of AE linear localisation.

**Figure 5 materials-16-03527-f005:**
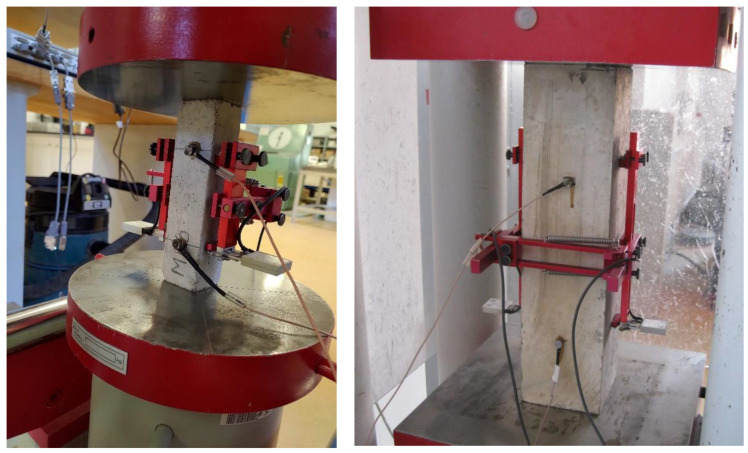
Location of acoustic emission sensors on a smaller (**left**) and bigger (**right**) specimen.

**Figure 6 materials-16-03527-f006:**
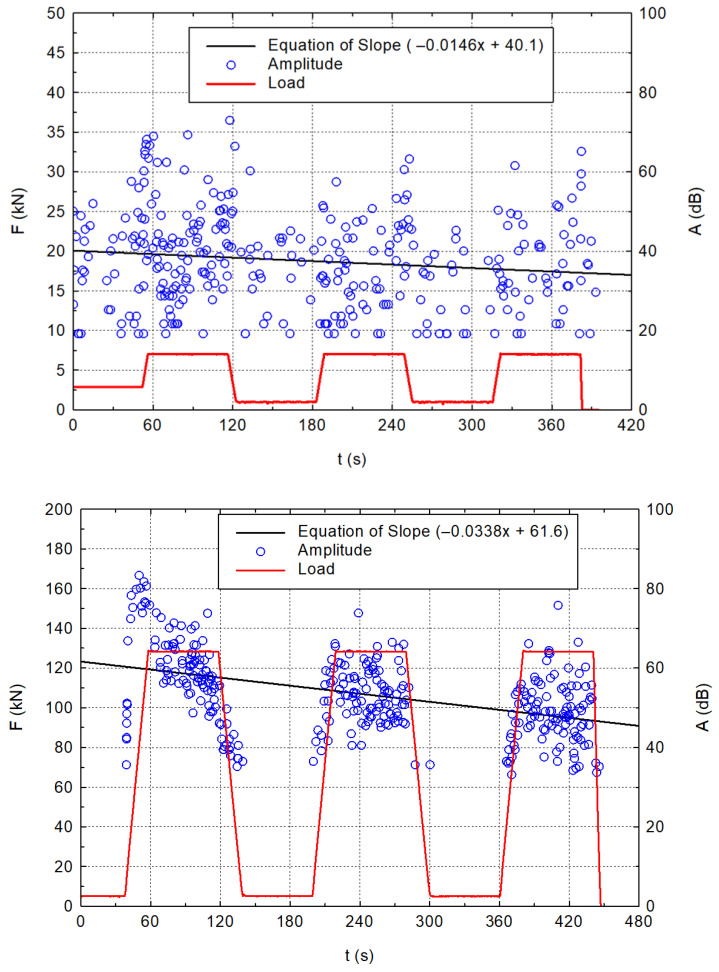
Example of the loading specimens during static modulus of elasticity with the recording of acoustic emission (amplitude of hits) for a smaller (**top graph**) and a bigger (**bottom graph**) specimen made of CEM III/A immersed in the acetic acid solution.

**Figure 7 materials-16-03527-f007:**
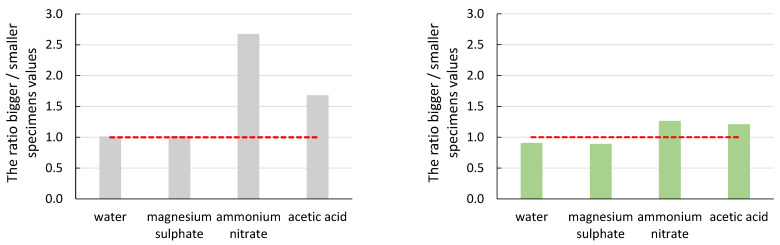
Graphical representation of the size effect on the static modulus of elasticity (E_c_).

**Figure 8 materials-16-03527-f008:**
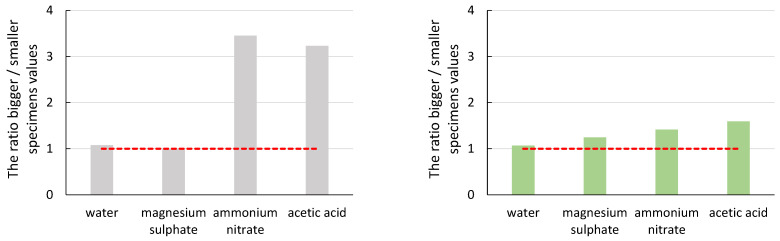
Graphical representation of the size effect on the compressive strength (f_c,prism_).

**Figure 9 materials-16-03527-f009:**
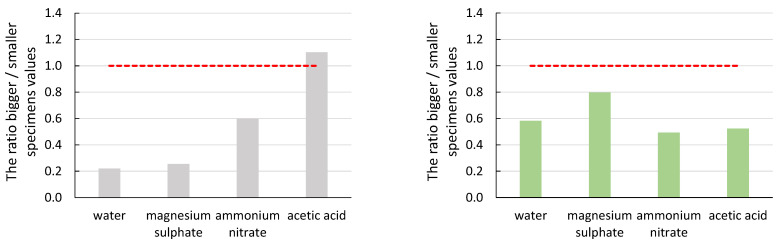
Graphical representation of the size effect on the number of acoustic emission hits.

**Figure 10 materials-16-03527-f010:**
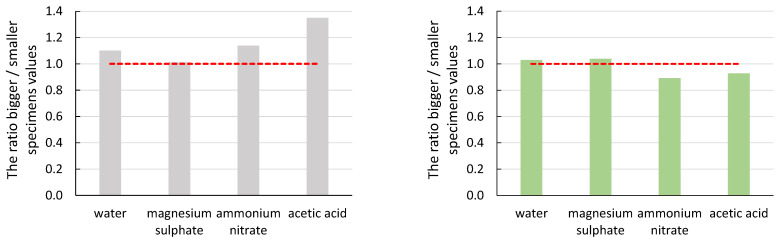
Graphical representation of the size effect on the value of the average amplitude of AE hits.

**Figure 11 materials-16-03527-f011:**
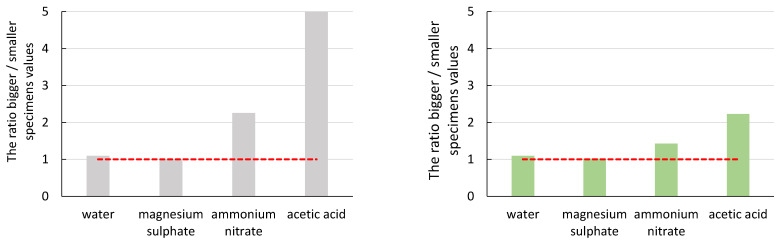
Graphical representation of the size effect on a slope of a line amplitude versus time.

**Figure 12 materials-16-03527-f012:**
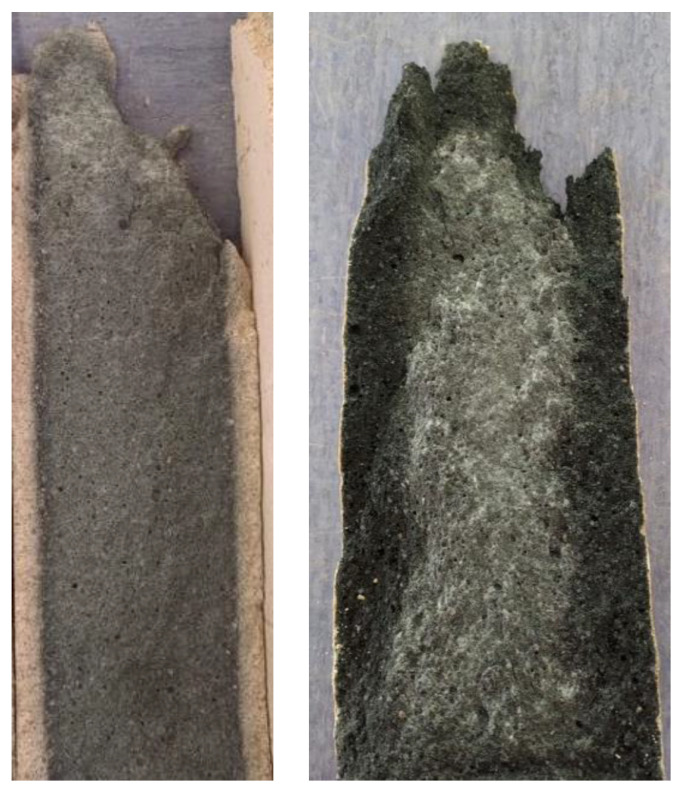
Comparison of degradation depth for both mortars in ammonium nitrate (**left**—CEM mortar, **right**—AAS mortar).

**Figure 13 materials-16-03527-f013:**
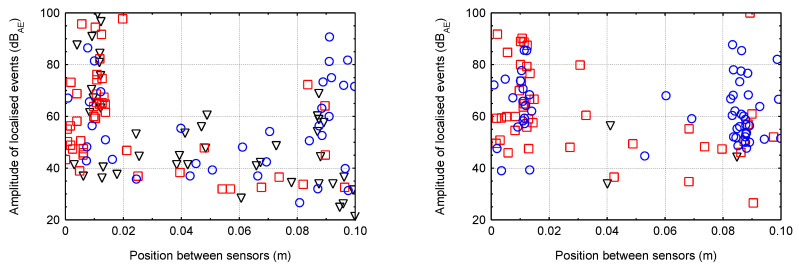
The localised events for the three specimens of CEM mortar (**left**—water; **right**—magnesium sulphate).

**Figure 14 materials-16-03527-f014:**
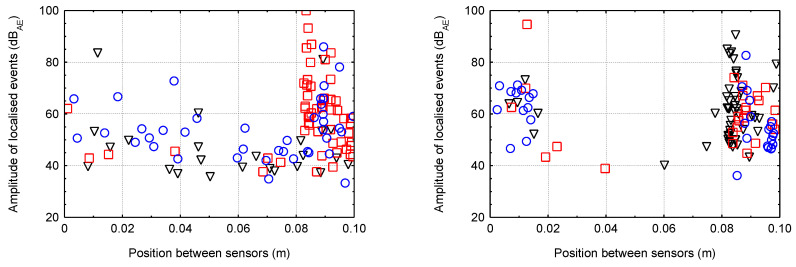
The localised events for the three specimens of AAS mortar (**left**—water; **right**—magnesium sulphate).

**Figure 15 materials-16-03527-f015:**
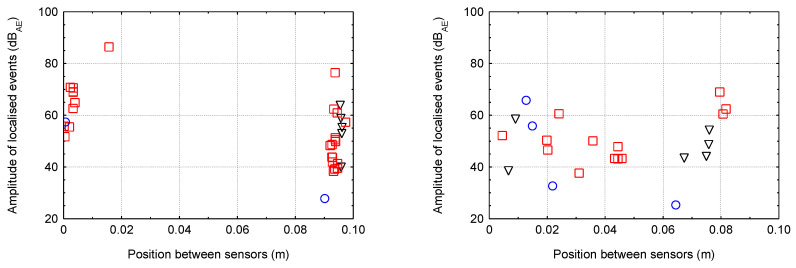
The localised events for the three specimens of CEM mortar (**left**—ammonium nitrate; **right**—acetic acid).

**Figure 16 materials-16-03527-f016:**
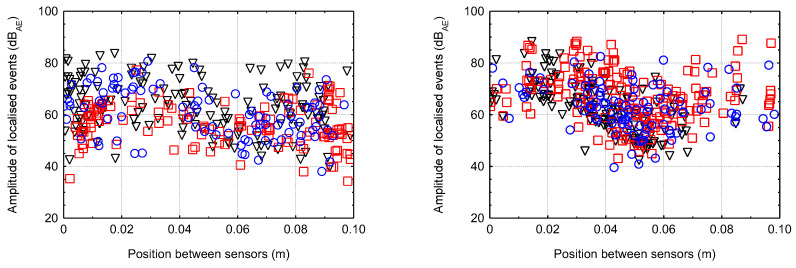
The localised events for the three specimens of AAS mortar (**left**—ammonium nitrate; **right**—acetic acid).

**Table 1 materials-16-03527-t001:** Chemical compositions (%) of Blast Furnace Slag (BFS) and Blast Furnace Cement (BFC = CEM III/A) as determined by X-ray fluorescence.

	CaO	SiO_2_	MgO	Al_2_O_3_	SO_3_	TiO_2_	K_2_O	MnO	Na_2_O	Fe_2_O_3_	LOI
BFS	37.0	39.4	8.6	8.1	1.4	0.3	1.2	0.9	0.4	0.7	2
BFC	45.6	31.6	5.8	7.4	3.3	0.4	0.8	0.6	0.3	1.4	2.8

**Table 2 materials-16-03527-t002:** Composition of individual mortar mixes (weight in grams).

Material	Blast Furnace Slag	CEM III/A	Sand	Sodium Water Glass (Ms = 0.5)	Water
AAS mortar *	22.0	–	66.1	4.6	7.2
CEM mortar **	–	22.5	67.4	–	10.1

* AAS mortar is made based on alkali-activated blast furnace slag (AAS) binder. ** CEM mortar is made based on blast furnace cement (CEM) binder.

**Table 3 materials-16-03527-t003:** pH value of solutions measured just before exchange for fresh solutions.

	Water		Acetic Acid		Magnesium Sulphate		Ammonium Nitrate	
Days	AAS	CEM	AAS	CEM	AAS	CEM	AAS	CEM
0	7.45		3		8.1		6.24	
28	12.48	10.78	4.99	5.1	9.25	9.67	9.37	9.64
56	12.65	10.57	5.37	5.49	9.39	9.53	9.57	9.71
84	9.45	7.73	4.52	4.92	8.89	9.47	9.02	9.71
112	9.66	8.53	4.57	4.89	8.87	9.3	9.11	9.13
140	9.03	8.33	4.5	4.64	8.9	9.06	8.69	8.13
168	8.35	8.05	4.24	4.17	8.58	8.91	8.58	8.66

**Table 4 materials-16-03527-t004:** Test results for CEM III/A-based mortar (values are the average of three specimens with coefficient of variation).

Specimen		E_c_ (GPa)	f_c,prism_ (Mpa)	Number of Hits (-)	Average AE Amplitude (dB)	Slope (dB/s)
Smaller	Water	40.56 (0.009)	61.85 (0.103)	1443 (0.010)	41.9 (0.153)	−0.027 (0.295)
	Magnesium sulphate	39.68 (0.010)	65.15 (0.069)	1213 (0.113)	46.8 (0.044)	−0.037 (0.104)
	Ammonium nitrate	9.44 (0.081)	10.78 (0.112)	262 (0.257)	39.4 (0.047)	−0.012 (0.212)
	Acetic acid	18.17 (0.052)	12.37 (0.185)	263 (0.250)	36.4 (0.074)	−0.005 (0.163)
Bigger	Water	40.99 (0.010)	66.55 (0.037)	318 (0.204)	46.1 (0.024)	−0.030 (0.157)
	Magnesium sulphate	40.40 (0.010)	64.89 (0.017)	310 (0.299)	47.4 (0.019)	−0.036 (0.078)
	Ammonium nitrate	25.26 (0.024)	37.21 (0.126)	157 (0.441)	44.9 (0.056)	−0.028 (0.374)
	Acetic acid	30.54 (0.014)	39.96 (0.017)	291 (0.231)	49.1 (0.106)	−0.031 (0.158)

**Table 5 materials-16-03527-t005:** Test results for alkali-activated slag-based mortar (values are the average of three specimens with coefficient of variation).

Specimen		E_c_ (Gpa)	f_c,prism_ (Mpa)	Number of Hits (-)	Average AE Amplitude (dB)	Slope (dB/s)
Smaller	Water	33.81 (0.043)	52.11 (0.093)	674 (0.111)	47.4 (0.039)	−0.026 (0.220)
	Magnesium sulphate	32.51 (0.038)	41.96 (0.078)	577 (0.018)	49.1 (0.022)	−0.042 (0.142)
	Ammonium nitrate	19.64 (0.046)	32.82 (0.090)	812 (0.154)	53.6 (0.069)	−0.022 (0.298)
	Acetic acid	19.41 (0.040)	29.08 (0.236)	1049 (0.044)	55.1 (0.048)	−0.018 (0.031)
Bigger	Water	30.64 (0.006)	55.61 (0.034)	393 (0.180)	48.8 (0.035)	−0.029 (0.151)
	Magnesium sulphate	29.02 (0.015)	52.19 (0.021)	460 (0.201)	51.0 (0.022)	−0.043 (0.098)
	Ammonium nitrate	24.82 (0.023)	46.47 (0.037)	400 (0.152)	47.8 (0.020)	−0.032 (0.171)
	Acetic acid	23.45 (0.051)	46.30 (0.016)	549 (0.173)	51.1 (0.022)	−0.041 (0.184)

## Data Availability

Data are available upon official request from the authors.

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
