# Peer review of "The Effect of Specimen Size on Acoustic Emission Parameters and Approximate Position of Defects Obtained during Destructive Testing of Cementitious and Alkali-Activated Degraded Fine-Grained Materials"

_materials, 2023, doi:10.3390/ma16093527_

Round 1

Reviewer 1 Report

In this paper, the effect of size on the degradation level of cementitious and alkali-activated degraded fine-grained materials was investigated. I recommend the publication of this manuscript after minor revision.

1) Line 34 and 36 the subscript in CO2

2) Line 37 4.4?

3) Line 44 What does “moment” refer to?

4) Line 59 “Blast Furnace Slag”  Is uppercase letters necessary?

5) Line 89-91 The superscript and subscript in the unit and CaF2.

6) Three-line tables for the format.

7) Line 266 Table 4?

8) No Y-axis can be seen in Figs. 6, 7, 9, 10 and 11.

9) It’s better to introduce “Kaiser effect” when it was firstly mentioned.

10) Line 348 “indicating minimal or no degradation due to degradation”?

11) What do the symbols refer to in Figs. 12-15?

12) The section of “Conclusion” should be summarized and simplified.

Author Response

Hello, dear reviewer
Thank you for your helpful review. We have included all your comments and suggestions for improvements in our article.

We hope you will be satisfied.

Collective of authors

Reviewer 2 Report

The authors investigate the size effect on acoustic parameters for cementitous and alkali aciveted fine grained materilas.. The paper is generally good but it needs improvement. Followings should be carried out before acceptance:

The abstract should contain important results of the study.

Remove We in line 24 Convert to passive sentence

Following studies are adviced to add first paragraph of introduction related to reduce CO2 : influence of replacing cement with waste glass on mechanical properties of concrete; use of recycled coal bottom ash in reinforced concrete beams as replacement for aggregate; concrete containing waste glass as an environmentally friendly aggregate: a review on fresh and mechanical characteristics; mechanical behavior of crushed waste glass as replacement of aggregates;flexural behavior of reinforced concrete beams using waste marble powder towards application of sustainable concrete; Production of perlite-based-aerated geopolymer using hydrogen peroxide as eco-friendly material for energy-efficient buildings; geopolymer concrete with high strength, workability and setting time using recycled steel wires and basalt powder.

Research significance/novelty is poorly written. The authors must improve this.

The reason for selecting design mixture should be added.

Compare your results with existing studies

Add also large specimen setup in Fig 4.

Add photos for utilized materials. There is no photo related to which materials are utilized.

Add recent studies on this subject to introduction. There are many studies on the introduction for this topic.

Conclusion should be improved. The recommendation consdiering all test should be given for engineers.

Author Response

(The authors gave the same response as above.)

Round 2

Reviewer 2 Report

The reviewer thanks to authors for improving the paper.

The paper can be acceptable.